# Radiolabeled Dendrimer Coated Nanoparticles for Radionuclide Imaging and Therapy: A Systematic Review

**DOI:** 10.3390/pharmaceutics15030867

**Published:** 2023-03-07

**Authors:** Miriam Conte, Maria Silvia De Feo, Marko Magdi Abdou Sidrak, Ferdinando Corica, Joana Gorica, Luca Filippi, Orazio Schillaci, Giuseppe De Vincentis, Viviana Frantellizzi

**Affiliations:** 1Department of Radiological Sciences, Oncology and Anatomo-Pathology, Sapienza, University of Rome, 00161 Rome, Italy; 2Department of Nuclear Medicine, Santa Maria Goretti Hospital, 04100 Latina, Italy; 3Department of Biomedicine and Prevention, University Tor Vergata, 00133 Rome, Italy

**Keywords:** dendrimer, theragnostic, oncology, nanoparticles, radiolabeled nanoprobe

## Abstract

Background: Dendrimers are nanoscale-size polymers with a globular structure. They are composed of an internal core and branching dendrons with surface active groups which can be functionalized for medical applications. Different complexes have been developed for imaging and therapeutic purposes. This systematic review aims to summarize the development of newer dendrimers for oncological applications in nuclear medicine. Methods: An online literature search was conducted on Pubmed, Scopus, Medline, Cochrane Library, and Web Of Science databases selecting published studies from January 1999 to December 2022. The accepted studies considered the synthesis of dendrimer complexes for oncological nuclear medicine imaging and therapy. Results: 111 articles were identified; 69 articles were excluded because they did not satisfy the selection criteria. Thus, nine duplicate records were removed. The remaining 33 articles were included and selected for quality assessment. Conclusion: Nanomedicine has led researchers to create novel nanocarriers with high affinity for the target. Dendrimers represent feasible imaging probes and therapeutic agents since, through the functionalization of external chemical groups and thanks to the possibility to carry pharmaceuticals, it can be possible to exploit different therapeutic strategies and develop a useful weapon for oncological treatments.

## 1. Introduction

The advent of nanomedicine has raised the necessity to develop better techniques to guarantee high specificity and sensibility in delivering drugs and in diagnosis, reducing signal-to-noise ratio in images [1]. In this regard, new Smart Drug Delivery Systems (SDDSs), delivery systems for pharmaceuticals that are based on smart nanocarriers, have been created [2]. These include the dendrimers which are nanoscale-size polymers with a globular structure and many branches, similar to a suction ball, constituting a core, the branching dendrons, and active groups of the surface [3]. They can be hydrophilic or hydrophobic on the basis of the physiochemical characteristics of surface-active groups. For their dimensions, as well as their low polydispersity index and the possibility of surface functionalization, they are employed as drug carriers likewise in imaging. As their name suggests (from the Greek “dendron” which means “tree/branch” and ‘meros’ which stands for “part”), dendrimers have a tridimensional structure in which all the bonds are arranged radially surrounding the core [4]. The bonds are repeated in units that provide branching points which give rise to a generation (G1, G2…) [5,6]. Between these bonds, interior generations and exterior generations can be distinguished (See Figure 1). The interior generations are the repeated units bonded to the core, the exterior generations are linked to the most external interior bonds and is are responsible for terminal functionality. Since dendrimers have a quick clearance by immune cells and this affects the uptake by cancer cells, different chemical modification has been studied such as copolymerization with linear polymers, hybridization with nanocarriers, and dendritic structure modifications or also exposing the surface to light, heat, enzyme activity, and modifications of ph, which permitted obtaining more specific targeting [7]. Moreover, a variety of hybrid compounds characterized by a metallic core and dendrimer shell have been proposed as delivery agents. The advantage of dendrimer-coated magnetic nanoparticles seems to be related to the possibility of combining the advantages of superparamagnetic iron oxide nanoparticles (SPION) with the versatility of dendrimers, in which cargo non-covalent encapsulation is possible as well functionalization of shells [1]. SPION themselves are crystals of iron oxide (maghemite or magnetite) with a shell that can be functionalized to obtain more stability in fluids and more linking specificity for a specific epitope. In addition, the sensitivity of the magnetic field allows the compound to be a useful contrast agent in magnetic resonance imaging (MRI) [8]. This systematic review aims to summarize the application and the development of the new dendrimer-coated nanoparticles which find applications in oncological nuclear medicine imaging and therapy.

## 2. Materials and Methods

### 2.1. Search Strategy and Study Selection

This systematic review respects PRISMA guidelines [9]. The consulted databases used in the online literature search were Pubmed, Scopus, Medline, Central (Cochrane Library), and Web of Science databases. Papers published from January 1999 to December 2022 were searched. The chosen keywords applied in each database research were: “dendrimer” AND “radiolabeled” AND “nanoparticles”. Eligible studies considered the application and development of radiolabeled dendrimeric nanomaterials for nuclear medicine imaging and therapy only in the oncology field. Reviews, studies not related to oncology, or not English language texts were not selected.

### 2.2. Quality of the Selected Studies

For each paper, general data such as the journal, authors, publication year, country, and study design were collected. The accepted studies were examined with the Quality Assessment of Diagnostic Accuracy Studies-2 (QUADAS2) tool. Data extraction and quality assessment were conducted by two reviewers independently. Any divergences were resolved by discussion among researchers.

## 3. Results

### 3.1. Search Results

The research produced a total of 110 articles; 78 articles were identified from PubMed, and 32 from Scopus, while no studies were detected in Cochrane library, Medline, and Web of Science. A total of 111 articles were reviewed through the examination of each abstract to select studies with potential relevance. From the overall group, 69 articles were excluded because they did not satisfy the inclusion criteria. Among them, 24 were reviews, three were books, one paper was an editorial, one was not in the English language and 40 were not related to the research field. Thus, nine duplicate records were removed. The remaining 33 articles were included and selected for quality assessment. The search strategy and applied selection criteria are represented in the flow chart (See Figure 2).

### 3.2. Study Characteristics

The main thematic areas of the selected studies could be summarized as follows: (1) studies of new dendrimeric radiopharmaceuticals developed for oncological nuclear medicine imaging, (2) dendrimeric radiopharmaceuticals for oncological nuclear medicine therapy, (3) in vitro studies of new dendrimeric complexes. In the first category, a relevant heterogeneity was found among the different studies concerning the various technologies employed for detection: both positron emission tomography/computed tomography (PET/CT) and single photon emission computed tomography (SPECT) were employed in the field. Concerning the different tumors investigated in the selected manuscripts, three studies were concerned with applications in melanoma, five on breast cancer, two on glioma, one prostate cancer, one dealt with application on apoptotic cells, seven on folate receptor positive cancer, one study considered Chlorambucil-complexed dendrimers, one study was on neuroendocrine tumors, two on detection of sentinel lymph nodes, one on Ehrlich’s ascites tumor, two on colon carcinoma, one on dendrimers applied on HeLa cells, one on fibrosarcoma, one on ovary cancer, one on angiogenesis application. Two studies were synthesis studies. Three studies analyzed dendrimers’ therapeutic features in detail, of which one was already cited in the group that considered folate receptor-positive cancers and one in the glioma group.

### 3.3. Methodological Quality Assessment

The methodological quality of the papers included studies of very high quality. All the selected papers satisfied all the QUADAS-2 domains and all studies obtained a low concern of bias (Figure 3).

## 4. Discussion

### 4.1. Synthesis Studies

The synthesis of a dendrimer is a stepwise procedure that could be conducted in two different ways: divergent or convergent method. The convergent synthesis consists of a separate generation of dendrons which are subsequently linked to the core. In the divergent approach, the dendrons are created outwards starting from the core. The final compounds are monodisperse in size, such as the approaches used for solid-phase polypeptide/oligonucleotide synthesis. This is a preferable result since it guarantees reproducibility experiments and therapeutic effects. Monodisperse compounds can be easily produced for low-generation dendrimers such as G3, but occasionally at higher generations, the inability to separate perfect dendrimers from similar dendrimers with minor flaws leads to a deviation from absolute monodispersity [10]. In biological applications, polyamidoamines [10], polyamines [11], polyamides (polypeptides) [12], poly(aryl ethers) [13], polyesters [14], carbohydrates [15], and DNA [16] are some of the dendrimer types that are frequently used, but the most exploited and available scaffold is the polyamidoamine (PAMAM).

McNelles and colleagues functionalized the core of generation 5 aliphatic polyester dendrimers with a dipicolylamine Tc (I) chelate and the periphery with vinyl groups. The compound was PEGylated with three distinct molecular weights mPEGs, mPEG160, mPEG350, and mPEG750 [17]. The dendrimer conjugated with PEG750 was demonstrated to be molecularly dispersed in water and was radiolabeled with [Technetium-99m (^99m^Tc) (CO)_3_(H_2_O)_3_]^+^. SPECT images were acquired at 30 min, 2 h, and 22 h after tail injection in two anesthetized male Copenhagen rats. A great part of the activity was concentrated in the blood, the lungs, and the heart which remains at 22 h, and a minor part in the kidneys. SPECT imaging was also conducted on three CD1 nude mice with subcutaneous squamous cell (H520) tumors at 2 and 6 h post-injection of [^99m^TcDPA-G5-(PEG750)]^+^. After 2 h only little tumor uptake was seen, but it became evident at 6 h after the tracer administration. 

However, the coating with PEG, even if it limits degradation and permits long-term use compared to natural proteins, can be affected by immune system activity which can reduce in vivo stability [18]. Hlídková et al., therefore, started from the synthesis of a monodisperse magnetic poly(glycidyl methacrylate) microspheres with amino groups (mgt.PGMA-NH_2_) through the multistep swelling polymerization of glycidyl methacrylate (GMA) [19]. The resulting microspheres were covered by three generations of an amino acid dendritic system (Ser-Lys-Ser/Lys-Ser/Lys-Ser) to inhibit undesirable non-specific protein adsorption from biological fluids and alkaline groups were added to the complex. In fact, they considered that Ser made dendrimer chains longer, which had a good impact on hydrophilization, while Lys facilitated the branching. The mgt.PGMA-D3 particles (0.2 g) containing residual NH_2_ groups were linked to alkyne groups and a “click reaction” with ^125^I-N_3_-RGDS peptide was conducted. Alkyne containing dendrimeric mgt. PGMA microspheres were demonstrated to be a feasible probe for fluorescence spectroscopy and a good carrier for drugs, antibodies, and functional groups useful for medical applications.

### 4.2. Oncological Applications

#### 4.2.1. Dendrimers and Melanoma

New melanoma cases were 96,445 in the USA in 2020, with an incidence rate of 3.4 between all ages and both sexes [20]. To ameliorate overall survival and the chance of treatments, several step-forwards have been conducted. An original approach is applying functionalized dendrimers for imaging in nuclear medicine.

In an innovative study, Li et al. developed a Gallium-68 labeled 2-amino-2 deoxy-D-glucose (DG) functionalized dendrimer (generation 5 polyamidoamine) conjugated with gold nanoparticles (Au DENPs) and cytosine-guanine (CpG) oligonucleotide for PET/CT scan [21]. The resulting complex (DG-Au-DENPs/CpG) was demonstrated to be a good imaging probe in the melanoma murine model and a good therapeutic agent in the mouse model of subcutaneous melanoma. The high performance and spatial resolution of PET/CT [22] gave the opportunity to obtain a sensitive tool. 

Another paper that studied the application of dendrimers for the imaging of melanoma is that of Tassano et al. They valued in a murine model of melanoma the biodistribution of poly(amido)-amine (PAMAM) generation 4 (G4) dendrimer labeled with ^99m^Tc [23]. High tumor uptake was seen at scintigraphy while a cytoplasmatic distribution was confirmed with confocal microscopy. A further report on the melanoma application of dendrimers is the work of Tanaka and colleagues [24]. They developed a polyamidoamine dendrimer (fourth generation; G4) conjugated with γ-PGA complex and radiolabeled with Indium-111 (^111^In)-labeled and Iodine-125 (^125^I)-labeled polyethyleneimine (PEI) for imaging of metastatic melanoma. In biodistribution studies executed on mice bearing B16-F10-lung metastatic cancer, selective tumor uptake was esteemed with a major lung-to-blood ratio of radioactivity than that of normal control. These results represent valid progress in diagnosis due to the noninvasivity of the methods and the frequent metastasis of melanoma at the early phase that requires a rapid clinical work-up. However, the different biodistributions, absorbed dose of critical organs which is influenced also by dendrimer functionalization, but also physical and biological half-life of ^99m^Tc, ^125^I and ^111^In could open a discussion in terms of radioprotection for further applications on humans. 

#### 4.2.2. Breast Cancer

Breast cancer is the most common cancer in females with 24.5% of new cases in 2020 between all cancer types, after the classified “other cancers” (37.8%) not including thyroid (4.9%), lung (8.4%), corpus uteri (4.5%), and colorectum cancer (9.4%) [20]. Five different studies examined novel dendrimeric complexes for breast cancer detection and treatment.

Ebrahimi et colleagues developed a second generation of citric dendrimer (dendrimer-G2) combined with a VEGF antagonist for imaging and therapeutic purposes in breast cancer exploiting the VEGFR overexpression in this malignancy [25]. They radiolabeled the compound with technetium-99m (^99m^Tc) tested in normal cells, HEK-293, and cancerous cells, and 4 T1 cell lines by MTT assay at different concentrations (10, 50, 100, 200, 500, 1000 µg/mL). After 24 h no toxicity was found in normal cells, while dose-dependent toxicity was seen in cancer cells, with 50% of 4 T1 cells dying at the concentration of 200 µg/mL. SPECT images were obtained from mice bearing 4 T1 lines demonstrating high uptake of ^99m^Tc-dendrimer-anti-VEGF in the tumor and, due to peptide conjugation, in the liver. These results highlighted the efficacy of this compound for breast cancer imaging and a feasible tool for prospective therapeutic purposes. To create multi-functionalized NPs, polyamidoamine (PAMAM) dendrimers were linked with diethylenetriaminepentaacetic acid (DTPA) and developed with PEG2000. Gefitinib (GEF) was placed into DTPA-PAM-PEG NPs, which were then equipped with the MUC-1 aptamer to produce the DTPA-PAM-PEG/GEF@MUC-1 nanosystem. The compound was radiolabeled with Gallium-67. SPECT images obtained from breast tumor-bearing rats showed a significant uptake of the tracer in the cancer site, indicating how the nanosystem is a promising image-guided tool for breast cancer that express mucin [26]. Zamani et al. produced a dendrimer-G3 linked to folic acid, overexpressed in breast cancer, conjugated with poly(ethylene glycol) (PEG)-citrate and radiolabeled with ^99m^Tc for breast cancer imaging [27]. Toxicity tests on HEK-293 cells demonstrated no considerable toxicity also at the highest concentration (8 mg/µL). SPECT imaging on breast cancer-bearing mice indicated a higher uptake in tumors at 60 min post-injection (Figure 4). Song et al., indeed, created ^131^I-labeled dendrimers modified with the LyP-1 peptide for cancer imaging and oncological therapy [28]. They modified the amine-terminated generation 5 poly(amidoamine) dendrimers with 33.1 LyP-1 peptide, capable to identify the overexpression of p32 in cancerous cells and able to internalize in 4T1 cells breast cancer as demonstrated in the conducted in vivo and in vitro SPECT evaluations. The compound was, therefore, conjugated with 9.2 3-(4′-hydroxyphenyl)propionic acid-OSu (HPAO). Then acetylation of the dendrimer terminal amines was conducted. ^131^I was used as a radioisotope for radiolabeling. The ^131^I-labeled LyP-1-modified dendrimers were demonstrated to be a good probe for SPECT imaging, a selective therapeutic tool both in vitro in 4T1 cells and in vivo in the 4T1 subcutaneous tumor mice model. Gibbens-Bandala et al. synthesized ^177^Lu-labeled polyamidoamine (PAMAM) dendrimer (DN) conjugated with paclitaxel (PTX) and surface functionalized with the Lys1Lys3 (DOTA)-bombesin (BN) peptide for selective targeting to gastrin-releasing peptide receptors (GRPr) which is overexpressed in breast cancer cells [29,30]. T47D Breast Epithelial Ductal Carcinoma cells positive for GRPr were used to establish the therapeutic effects in vivo. In particular, the compound was demonstrated to be more effective than PTX or ^177^Lu alone while the interaction between BN and GRPr guaranteed a selective uptake of cancer cells. SPECT images executed on a T47D subcutaneous tumor mouse model showed important tracer uptake in cancerous tissue and a tumor size reduction of 15.6% at 120 h after the injection.

These examples demonstrated how the use of dendrimers opens the door to supplementary possibilities in imaging but especially in therapy.

#### 4.2.3. Glioma Imaging

Zhao et al. synthesized new ^99m^Tc-labeled generation two dendrimer-entrapped gold nanoparticles (Au DENPs) conjugated with Diethylene Triamine Pentaacetic Acid (DTPA) and polyethylene glycol (PEG) modified with chemokine receptor-4 (CXCR4) ligand, FC131 peptide, based on the fact that CXCR4 is overexpressed in glioma cells and is associated with short survival [31]. The tracer showed radiochemical stability, and high affinity for CXCR4 expressing malignancies and was proposed for SPECT/computed tomography (CT) imaging of glioma and of different cancers with CXCR4 expression. Another study proposed a novel compound for glioma imaging and therapy [32]. Generation 5 amine-terminated poly(amidoamine) dendrimers were combined with polyethylene glycol (PEG), targeting agent chlorotoxin (CTX), and 3-(4′-hydroxyphenyl)propionic acid-OSu (HPAO). Then the acetylation of the terminal amines and ^131^I labeling was performed. Mice bearing C6 rat glioma cell line were injected with the resulting compound, iodine-131 (^131^I)-G5.NHAc-HPAO-(PEG-CTX)-(mPEG) dendrimers, underwent SPECT imaging. After 6, 15, and 24 h post-injection, higher tumor uptake was observed in mice treated with the CTX-targeted dendrimer compared to mice injected with the nontargeted dendrimer. 

This work defines an interesting approach: the availability of technetium ensures easy applicability of the procedure since not all the nuclear medicine centers all over the world have PET/CT instrumentation.

#### 4.2.4. Prostate Cancer

Lesniak et colleagues tested out the generation 4 PSMA-targeted polyamidoamine (PAMAM) dendrimers [G4(MP-KEU)] radiolabeled with ^64^Cu and ^111^In in prostate cancer (PCa) mice model [33]. They used a cell line with high expression of Prostate-specific membrane antigen (PSMA + PC3 PIP) and with the low expression one (PSMA − PC3 flu). Then they implanted the cells into mice and conducted PET/CT and SPECT/CT evaluations. Both ex vivo biodistribution analysis and SPECT/CT scans showed a selective accumulation of indium-111 (^111^In)G4(MP-KEU) in PSMA + PC3 PIP tumors compared to PSMA − PC3 flu cancers. PET/CT imaging demonstrated a modest tumor uptake of [^64^Cu]G4(MP-KEU) 1 h after the injection, but it increased after 24 h and confirmed high after 48 h while in vivo evaluation showed higher uptake in PSMA + PC3 PIP cells versus PSMA- ones. This result suggests that G4(MP-KEU) may be a suitable scaffold for imaging and therapeutic agents against PSMA-positive tumors. Given the diminished accumulation of (^111^In)G4(MP-KEU) in mice with low expression of PSMA compared to PSMA + PC3 PIP bearing mice, it should be interesting to compare the actual worldwide approved tracers such as ^68^Ga-PSMA, ^11^C-choline or ^18^F-PSMA-1007 with this novel radiopharmaceutical in terms of lymph-node detection also in correlation with prostate-specific antigen (PSA) hematic levels to understand if this tool could be a more specific and sensitive tracer, especially at low PSA levels.

#### 4.2.5. Dendrimer and Apoptotic Cells

Since anatomical alterations are later manifestations after metabolic changes, PET/CT and SPECT/CT are a cornerstone in therapeutic response. In fact, morphological approaches such as computed tomography can value the reduction in volume of the neoplasm but cannot precisely distinguish viable tumors from necrosis or fibrosis. Functional imaging can efficiently detect post-treatment tumor necrosis and predicts the tumor response [34].

In this context, the study of Xing et al. in which generation five PAMAM dendrimers (G5.NH2) were conjugated with 1,4,7,10-tetraazacyclododecane-1,4,7,10-tetraacetic acid (DOTA), polyethylene glycol (PEG) modified duramycin, PEG monomethyl ether, and fluorescein isothiocyanate (FI). Then gold nanoparticles were entrapped, and the compound was radiolabeled with ^99m^Tc [35,36]. They exploited the affinity of duramycin for phosphatidylethanolamine (PE), a protein that is expressed by cells during apoptosis, to develop a SPECT imaging tool for apoptotic cells. An in vitro and in vivo SPECT/CT scan was conducted, respectively, on apoptotic and normal C6 cells and subcutaneous apoptotic cancer cell mice models. Doxorubicin treatment was conducted for 3 days, consequently, SPECT images of ^99m^Tc-duramycin-Au DENPs treated cancers had selective tumor uptake compared to ^99m^Tc-Au DENPs treated cancers images (Figure 5). The in vivo images demonstrated high uptake in the liver and spleen, lower in the heart, lung, tumor, kidney, intestines, stomach, and soft tissue. The study assessed the utility of the compound for early identification of chemotherapy-induced apoptosis. The only possible limitation of this tool could be the hepatic and splenic tumor localization which in mice could be difficult to detect due to the biodistribution. However, biodistribution studies in humans to evaluate the possible limits of the method are needed.

#### 4.2.6. Folate Expressing Malignancies and Dendrimers

CD 44 is a transmembrane glycoprotein responsible of cellular proliferation, migration and invasion, so involved also in cancer growth, progression, metastasis and drug resistance [37] and so its expression is higher in several malignancies [38]. Hyaluronic acid (HA) is repeated disaccharide units of d-glucuronic acid and *N*-acetyl-d-glucosamine and is a component of the extracellular matrix, essential in cancerous invasion and metastasis. CD44 is the principal receptor of HA and so an important target for imaging and therapeutic aims [39,40]. 

HA carboxyl groups were complexed to the carboxyl groups of indium ^111^-labeled polyamidoamine (PAMAM) dendrimers and amino groups of polyethyleneimine to develop an imaging probe for CD44-positive cancers [41], exploiting the capacity of HA to link these types of tumors. Partial specific tumor uptake of indium 111-HA complex was observed in CD44 cancer-bearing mice.

The interesting use of the PAMAM dendrimer gives the possibility to conveniently conjugate the imaging agent thanks to the variety of active groups of the dendrons. This characteristic makes the compound useful also as anti-cancer drug carrier.

##### KB Cell Line and Dendrimer

The feasibility and the possibility to modulate the function of PAMAM leads different researchers to study it as a possible platform to carry imaging agents. An example is the study of Song et al. in which ^99m^Tc-labeled folate-polyamidoamine dendrimer was modified with 2-hydrazinonicotinic acid (^99m^Tc-HP3 FA) for imaging of folate receptor positive malignancies [42]. In the cellular uptake study, remarkable uptake was revealed into the folate receptor (FR) overexpressed KB cell line and blocked by an excess of folic acid, while no uptake was seen in the control group. Healthy BALB/c mice and mice bearing FR-positive KB cancer underwent SPECT imaging: in the first group, normal distribution was studied and the kidneys, rich in FR, were characterized by important uptake, while low levels were revealed in the liver and lung. In the second group, the tracer accumulated in the kidneys, and the cancer site localized on both shoulders. The tumor uptake was blocked by an excess of folic acid injected and no renal uptake was seen but only non-FR renal excretion was detected. In the research of Zhang et al. three conjugates of dendrimer PAMAM generation five, ^99m^Tc-G5-Ac-pegFADTPA, ^99m^Tc-G5-Ac-FA-DTPA, and ^99m^Tc-G5-Ac-DTPA, were radiolabeled with ^99m^Tc [43]. The ^99m^Tc labeled PEGylated dendrimer PAMAM folic acid conjugate (^99m^Tc-G5-Ac-pegFA-DTPA) compound demonstrated higher uptake in folate receptor positive KB cancer cells compared to the other two conjugates. In particular, the structure of PAMAM permitted obtaining an excellent pharmacokinetic profile. Through the functionalization of the compound such as replacing FA, for example, it is possible to obtain sensitive multimodal imaging or a therapeutic carrier for different oncological applications.

##### Lung Adenocarcinoma

In the paper of Ma et colleagues, copper-64 (^64^Cu)-labeled multifunctional dendrimers were proposed as PET tracers for the detection of folate receptor (FR)-expressing tumors [44]. They synthesized generation five of poly(amidoamine) dendrimers modified with fluorescein isothiocyanate (FI), folic acid (FA), and 1,4,7,10-tetraazacyclododecane-1,4,7,10-tetraacetic acid (DOTA). Figure 6 shows the schematic representation of DOTA-FA-FI-G5·NHAc dendrimers synthesis. The acetylation of the terminal amines was conducted, and the compound was radiolabeled with ^64^Cu through DOTA chelation. The cellular uptake was selective for the FR-positive KB cell line. Nude mice bearing KB cell line and A549 human lung adenocarcinoma cell line underwent a 5 min PET/CT scan acquired at 1,2,4 and 6 h post tracer injection, while a 10 min PET/CT scan was acquired 20 h post injection. The images showed a specific tumoral uptake and minimal uptake in the other organ at 20 h with the only exception being the liver, the kidneys, and the intestine.

Notably, the use of DOTA chelator permits the radiolabeling with other isotopes such as ^111^In, for SPECT imaging, but also ^90^Y and ^177^Lu for radiotherapy.

##### Breast Cancer

Mendoza-Nava et al. synthesized Lutetium-177 (^177^Lu)–dendrimer conjugated to folate and bombesin with gold nanoparticles in the dendritic cavity (^177^Lu–DenAuNP–folate–bombesin) in T47D breast cancer cells combining the photothermal and the radiotherapeutic properties of the nanoprobe [45]. After 6 min of laser irradiation, a considerable decrease in cellular viability was observed. The authors observed that hyperthermia makes cancer cells more sensitive to radiotherapy enhancing tumor oxygenation and meddling with the reparation of DNA but healthy tissue can be spared adding AuNP to dendrimer, and therefore, reaching a spatial precision for treatment. The same research group studied [46] ^177^Lu-dendrimer(PAMAM-G4)-folate-bombesin with gold nanoparticles (AuNPs) in the dendritic cavity as a theragnostic tracer of breast cancer expressing folate receptors (FRs) and gastrin-releasing peptide receptors (GRPRs). Preamble binding studies executed in T47D breast cancer cells demonstrated a specific cell uptake assessing the feasibility of this complex for nuclear imaging and targeted radiotherapy in GRPRs and FRs-positive breast cancer.

##### Generation Five Dendrimer Folic Acid

Another study based on folate receptors and the use of PAMAM is the evaluation of Cui et al., which tested a generation five polyamido amine (PAMAM) dendrimer folic acid conjugate radiolabeled with rhenium-188 (^188^Re) [47]. The research stated the high efficiency and stability of the compound in phosphate buffer and serum over 6 h after radio labeling. Chiefly, the use of PAMAM allowed a simply and efficient radiolabelling demonstrated to be stable in phosphate buffer saline. 

#### 4.2.7. Chlorambucil Complexed Dendrimers

In the study of Ghoreishi and colleagues, used a novel anionic linear globular PEG-Based Dendrimer-Chlorambucil radiolabeled with ^99m^Tc [48]. Biodistribution studies were conducted on rabbits after 50 min post injection through whole-body SPECT imaging. The nano-conjugate was filtered by the kidneys and excreted through the biliary system. However, the preponderant role was carried out by the urinary system, as demonstrated by images acquired at 24 and 48 h, underlying the good solubility of the compound due to the chemical properties of the citric acid-based-PEG dendrimer. The noteworthy uptake in the stomach at 24 h was due to to the biodegradable features of the tracer. Finally, they stated as the notable solubility of chlorambucil was improved by the nano compound which could be a good carrier for lipophilic pharmaceuticals.

Surely, the choice of PEG has been forward-looking thanks to the biocompatibility [49], the solubility [50] and the low toxicity of PEG compare to PAMAM [4] and the good capability to link with technetium creating a strong complex [51].

#### 4.2.8. Neuroendocrine Cancer

Two nanosystems were developed for neuroendocrine tumor imaging by Orcio-Rodríguez et al. ^99m^Tc-PAMAM dendrimer (G3.5)-Tyr3-Octreotide and ^99m^Tc-AuNP-Tyr-Octreotide were studied in somatostatin receptors (SR)-positive AR42J cancer cells and biodistribution evaluations were conducted in athymic mice bearing AR42J tumors [52]. Through a scintillation detector, selective tumor uptake of ^99m^Tc-PAMAM-Tyr3-Octreotide was seen 2 h after injection in the pancreas owing to high SR expression, but no difference in uptake profiles was found between ^99m^Tc-PAMAM-Tyr3-Octreotide and ^99m^Tc-AuNP-Tyr-Octreotide. The only difference was the higher renal elimination of ^99m^Tc-PAMAM-Tyr3-Octreotide. These results demonstrated the possible feasibility of these two tracers in neuroendocrine imaging. The only limit in the application of this method could be the pancreatic localization of a cancerous lesion but further evaluation in humans is needed.

#### 4.2.9. Application of Dendrimer in Lymph Node Detection

Niki et colleagues searched for the ideal dendrimer structure for sentinel lymph node (SLN) imaging. Twelve types of dendrimers of generation 2, 4, 6, and 8 and distinct terminal groups (amino, carboxyl, and acetyl) were synthesized and intradermally injected in rats’ right footpads. All G2 dendrimers primarily concentrated in the kidneys while amino-, acetyl-, and carboxyl-terminal dendrimers were primarily found at the injection site, in the blood, and in the SLN, respectively. In the SLN, macrophages, and T-cells mostly failed to identify the carboxyl-terminal dendrimers. Carboxyl-terminal dendrimers with generation 4 or higher single photon emission computed tomography imaging effectively detected SLN [53]. These results suggested how lower generations, especially carboxyl-terminal dendrimers, have to be taken carefully for SLN application, while G4 are more efficient in SLN detection. In particular, Sano et al. studied a ternary anionic complex formed by a polyamidoamine dendrimer (generation 4; G4) modified with diethylenetriamine pentaacetic acid (DTPA) derivative, polyethyleneimine (PEI), and γ-polyglutamic acid (PGA) and labeled with Indium-111 (^111^In). While cationic charge ^111^In-DTPA-G4/PEI was associated with high cytotoxicity, the anionic ^111^In-DTPA-G4/PEI/γ-PGA had no cytotoxicity. ^111^In-DTPA-G4/PEI/γ-PGA was characterized by a more relevant macrophage cell uptake than that of ^111^In-DTPA-G4/PEI. ^111^In-labeled nano-platforms were intradermally injected into rat footpads and the relevant radioactivity of ^111^In-DTPA-G4/PEI/γ-PGA was seen in the first draining popliteal LN; it was higher than that of ^111^In-DTPA-G4/PEI and was easily detected through single photon emission computed tomography (SPECT) [54]. ^111^In-DTPA-G4/PEI/γ-PGA could be a very efficient tool even more so the low cytotoxicity and hematotoxicity and seems able to overcome the limit of the negative charge of the surface responsible of electrostatically cellular repeal, suggesting an energy dependent uptake mechanism.

#### 4.2.10. Ehrlich’s Ascites Tumor

In the work of Ghai et al. etraazacyclododecane tetraacetic acid mono (N-hydroxysuccinimide ester) (DOTA-NHS active ester) was combined to generation 4 PAMAM dendrimers and radiolabeled with gallium-68 (^68^Ga) [55]. PET imaging conducted on BLAB/c mice bearing Ehrlich’s ascites tumor (EAT) cell lines showed a specific tumor uptake chiefly evident 1-h post the injection of the up cited nanoprobe.

As written previously, the use of DOTA chelator represents always a good choice since it permits a relevant stability and an efficient radiolabeling of the dendrimer.

#### 4.2.11. Colon Carcinoma

Albino BALB/c mice bearing human colon adenocarcinoma grade II cell line (HT29) underwent SPECT scintigraphy after the injection of two new nano analogs of methionine, Anionic Linear Globular Dendrimer G2, and DTPA-Methionine1 conjugates, both radiolabeled with ^99m^Tc. Tumor uptake was evident after 15 min post-injection, with higher resolution obtained with ^99m^Tc-Dendrimer-Met, demonstrating the feasibility of imaging and therapeutic uses of these two compounds [56]. The high purity of the complexes, the rapid tumor accumulation and the low uptake in the background make them interesting imaging agents. In the paper of Guillaudeu et colleagues, hydrophilic dendrimers with eight functional groups on their core were synthesized and labeled with iodine-125 (^125^I). Biodistribution studies were conducted on BALB/c mice and proved a long circulation half-life. The carboxylate version was coupled to a hydrazide linked, and to Doxorubicin. The nanocarrier carried a quantity of Doxorubicin in tumor site of C26 colon carcinoma cells-bearing mice equal to that carried by PEGylated liposomal formulation DOXIL, sparing more healthy tissue [57]. The advantage of this complex is the lack of accumulation in normal tissue, confirmed also after Doxorubicin conjugation, which permits accurately delivering the drug to cancer cells and maybe limiting possible side effects but further evaluations are required. 

#### 4.2.12. HeLa Cells In Vitro Study

A generation five of polyamidoamine (PAMAM) dendrimer (G5-Ac) was reacted with biotin and 2-(p-isothiocyanatobenzyl)-6-methyl-diethylenetria minepentaacetic acid (1B4M-DTPA), to create the compound Bt-G5-Ac-1B4M which was further complexed with avidin to form Av-G5-Ac-1B4M [58]. The last two complexes were labeled with ^99m^Tc and Av-G5-Ac-1B4M-^99m^Tc and showed significant cellular uptake in HeLa cells compared to Bt-G5-Ac-1B4M-^99m^Tc in an in vitro study. This study based on the evidence of higher internalization of avidin in ovarian and colorectal adenocarcinoma cells, which expresses b-D-galactose receptors [59]. The choice of radiolabelin with thecnetium, moreover, allows to obtain an easy to synthesize instrument thanks to the ease of obtaining the technetium. 

#### 4.2.13. Fibrosarcoma

Another report performed biodistribution studies of novel dendrimer complex in fibrosarcoma murine model [60]. They developed a generation 4 PAMAM dendrimer-coated MIONPs (IO@G4PM) modified with p-SCN-Bz-DOTA with the acetylation-free amine groups and radiolabeled with Gallium-68. IO@G4PM(DOTA3) (Ac) NPs as chosen candidates, IO@G4PM (DOTA3) and IO@G4PM (DOTA1) (Ac) NPs as controls were intravenously injected mice bearing fibrosarcoma. PET scans showed selective tumor uptake with a clearance rate lower for IO@G4PM (DOTA3) (Ac) NPs compared to IO@G4PM (DOTA3) and IO@G4PM (DOTA1) (Ac) NPs. Moreover, IO@G4PM (DOTA3) (Ac) samples (various Fe concentrations) were subjected to T2 relaxivity time measurement by a clinical 3 T MRI and a good linear correlation between the weighed T2 relaxation rate and concentration of iron was demonstrated. This complex is an exciting possibility to conjugate the functional evaluation with morpho structural images with the administration of a unique compound. 

#### 4.2.14. Ovarian Cancer

Sadekar et al. studied the biodistribution of linear N-(2-hydroxylpropyl)methacrylamide (HPMA) copolymers in comparison with that of branched poly(amido amine) dendrimers with surface hydroxyl groups (PAMAM-OH) labeled with ^125^I in murine orthotopic ovarian carcinoma [61]. PAMAM dendrimer hydroxyl terminated of generation 7.0 was characterized by longer plasma circulation, higher tumor uptake, and a major tumor-to-blood ratio. As was highlighted in the study, the architecture of a compound influences the increment in hydrodynamic radius and so the molecular weight which in turn affected the in vivo distribution, in particular renal and hepatic uptake. 

#### 4.2.15. Dendrimers and Tumor Angiogenesis and Metastasis

In the paper of Dijkgraaf et al., the production of a series of alpha (V) beta (3) integrin-directed monomeric, dimeric, and tetrameric cyclo[Arg-Gly-Asp-d-Phe-Lys] dendrimers through “click chemistry” was described. The compound was studied as an antagonist of α_ϒ_β_3_ integrin, a transmembrane protein with a notable role in tumor angiogenesis and metastasis [62]. A SK-RC-52 cell line was injected subcutaneously in nude BALB/c mice and ^111^In-labeled dendrimers 23, 24, and 25 compounds were evaluated for biodistribution studies which confirmed a selective uptake in tumors expressing α_ϒ_β_3_ integrin.

### 4.3. Therapeutic Applications of Dendrimers

Mamede et al. developed avidin (Av)-generation 4 dendrimer-chelate complex, labeled with indium-111, which through Auger and conversion electrons could be a feasible tool for internal radiation therapy [63]. They demonstrated an internalization of 75% after 24 h with a high tumor-to-background ratio in a mice model of an intraperitoneal tumor, with tolerable and dose-dependent therapeutic effects.

These results are innovative for the use of indium-111 which thanks to Auger electrons, the short path length and high LET resembles an alpha particle. However, the success of therapeutic effects could be reached only when the Auger emitter is carried near enough to the nucleus limiting the application to microscopic lesions.

## 5. Conclusions

The continuous necessity to develop more specific and simple tools for nanomedicine has led researchers to create novel nanocarriers useful for imaging and theragnostic purposes. Between them, dendrimers take a special seat for the capacity to carry pharmaceuticals to cancer cells with high specificity and for the possibility to enhance the binding affinity through the functionalization of external chemical groups. However, from the analysis of the scientific evidence, some final considerations can be made. First, dendrimer-based studies, although extremely promising, are at an embryonic phase and still far from passing from “bench to bedside”. In addition, we have registered the relevant heterogeneity in the technological devices employed for detection (SPECT or PET/CT) and this might have further influenced the findings reported in the selected papers. Despite the aforementioned limitations, innovative dendrimer-based approaches hold the promise to build a medicine in which a single tracer can behave as an imaging probe and therapeutic agent, combining different therapeutic strategies which could be a good alliance in the battle against cancer.

## Figures and Tables

**Figure 1 pharmaceutics-15-00867-f001:**
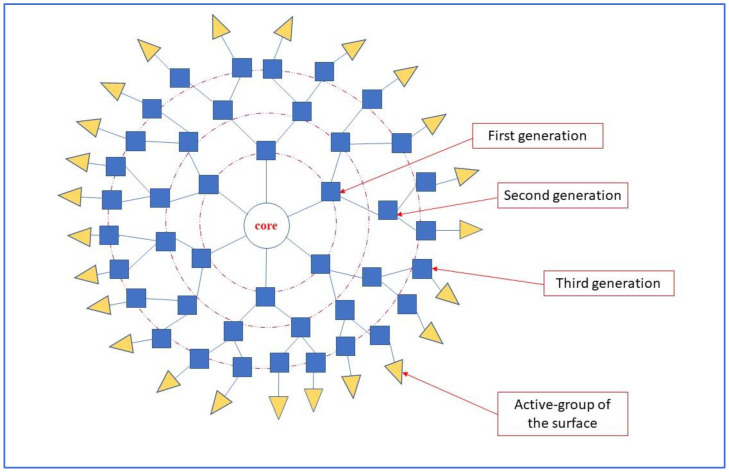
A graphic representation of a dendrimer. In red boxes the denomination of each component of the dendrimeric structure. As indicated, a central core and the radially-disposed dendrons, from the interior branches (first generation) to the external branches (third generation). The branch points divide the dendrons into generations. In the more external region, in yellow triangles, the functional groups.

**Figure 2 pharmaceutics-15-00867-f002:**
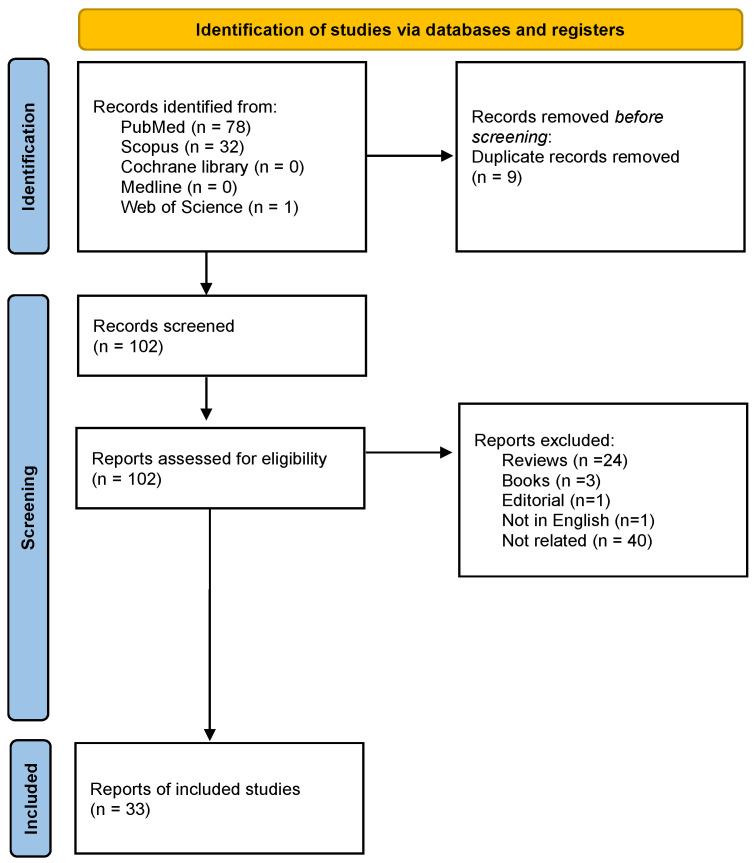
PRISMA 2020 flow diagram for new systematic reviews which included searches of databases, registers, and other sources.

**Figure 3 pharmaceutics-15-00867-f003:**
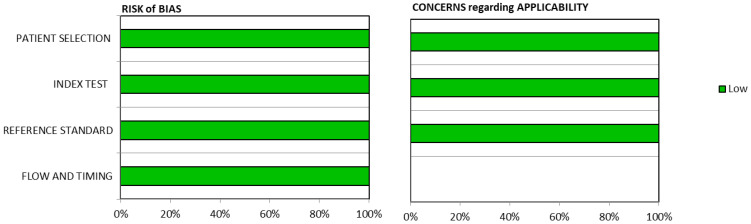
Graphic representation of the bias risk results assessed through QUADAS-2. All the selected articles have a low risk of bias and low concerns regarding applicability in all the domains as depicted by green bands.

**Figure 4 pharmaceutics-15-00867-f004:**
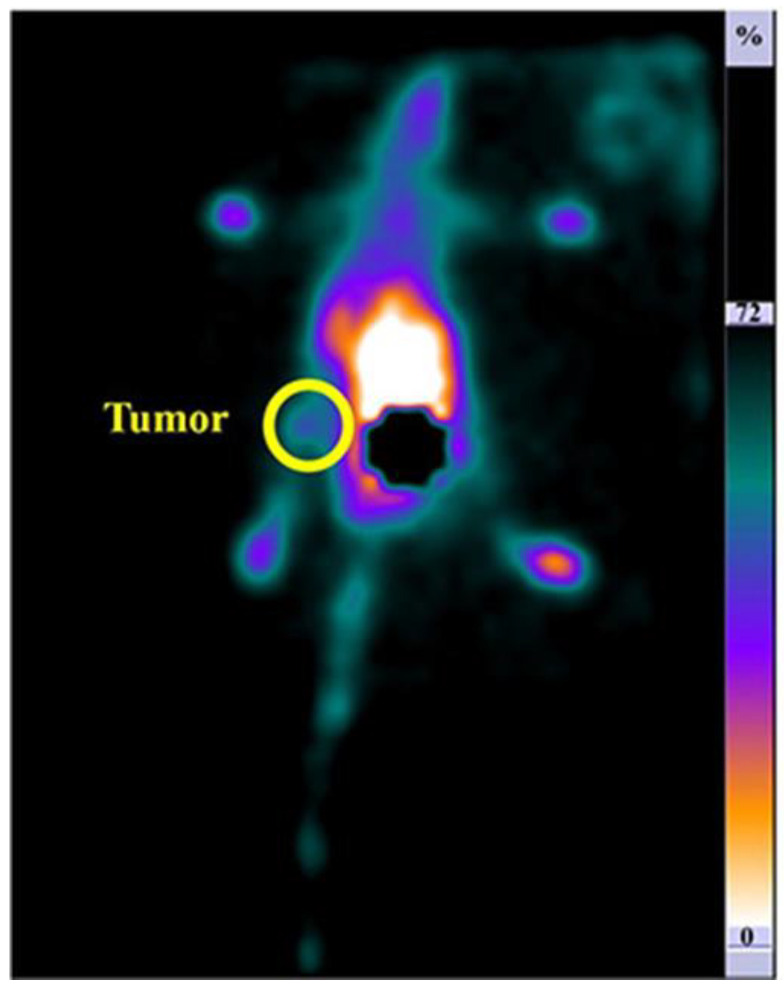
Planar scan (posterior revelation) taken from the work of by Zamani et al. in breast cancer-bearing mice 60 min after the injection of the ^99m^Tc—(PEG)-citrate dendrimer-G3 linked to folic acid. In the yellow circle, the tumor site.

**Figure 5 pharmaceutics-15-00867-f005:**
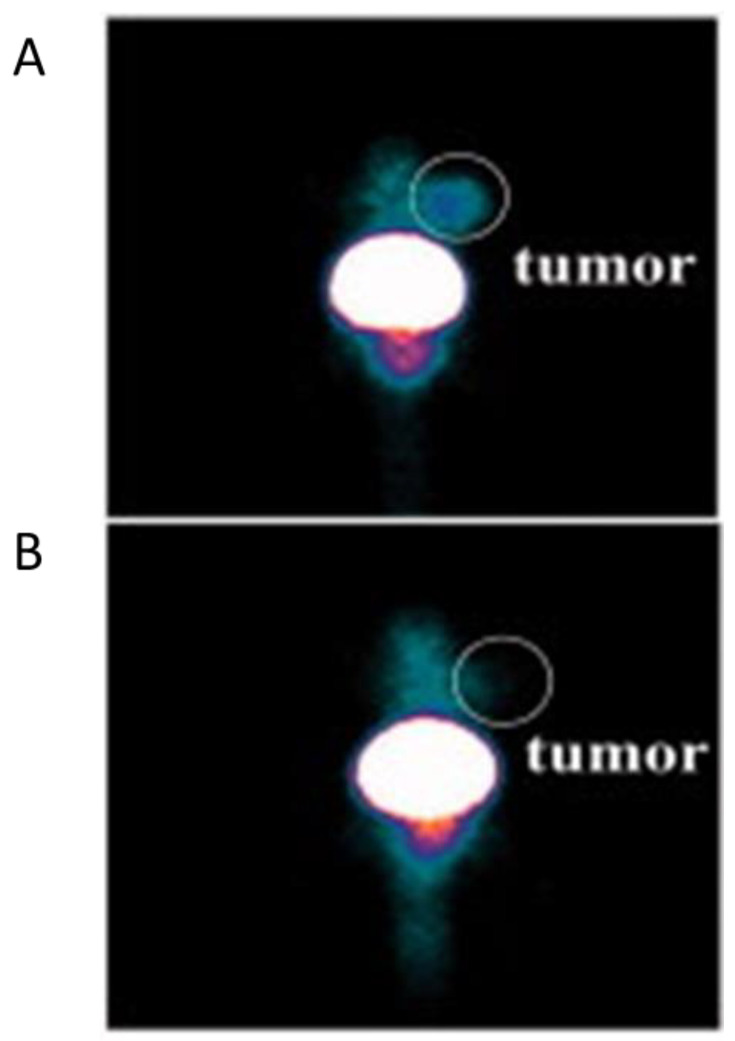
SPECT images of C6 xenograft tumor bearing mice after 3 days of DOX treatment at 8 h post tracer injection: in (**A**), mice were injected with ^9m^Tc-duramycin-Au DENPs, in (**B**) with ^99m^Tc-Au DENPs [35].

**Figure 6 pharmaceutics-15-00867-f006:**
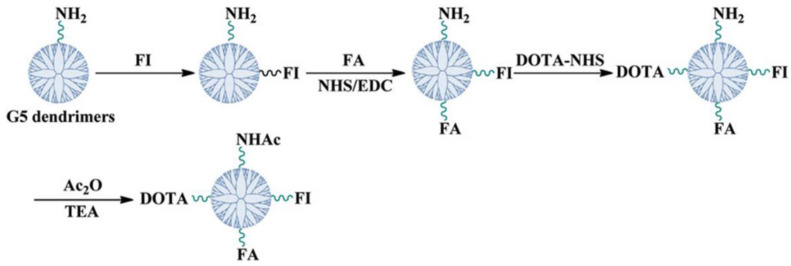
Schematic illustration of DOTA-FA-FI-G5·NHAc dendrimers synthesis [44].

## Data Availability

Not applicable.

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
