# Peer review of "Radiolabeled Dendrimer Coated Nanoparticles for Radionuclide Imaging and Therapy: A Systematic Review"

_pharmaceutics, 2023, doi:10.3390/pharmaceutics15030867_

Round 1
Reviewer 1 Report
Major remark: the title of review “Dendrimer-coated magnetic particles for radionuclide imaging and therapy: translational perspectives” does not correspond to the content.
Line 72: The chosen keywords applied in each database research were: “dendrimer” AND “radiolabeled” AND “nanoparticles”.
There are no "magnetic particles" included to the keywords search. Accordingly, magnetic particles are not the object of study in most of the works considered in the present review.
Line 73: Eligible studies should be considered the application and development of dendrimeric nanomaterials radiolabeled for nuclear medicine imaging and therapy only...
You must change either the title of the review or the search criteria and rebuild all of text body.
--------------------
Minor remarks:
Line 19: Methods: An online literature search was conducted on Pubmed, Scopus, Medline, Cochrane Library, and Web Of Science databases selecting published studies from January 2004 to December 2022.
Line 69: The consulted databases used for the online literature search were Pubmed, Scopus, Medline, Central (Cochrane Library), and Web of Science databases. Papers published from January 1999 to December 2022 were searched.
Does search begin from 1999 or 2004? Two statements need to be reconciled.
Line 126: Synthesis’ studies
The review should consider several examples of the dendrimer synthesis. Or at least describe the general scheme for the synthesis of such objects (radiolabeled dendrimers or dendrimer-coated magnetic nanoparticles) first, and then give the most relevant example(s).
Line 133: (CO)3(H2O)3]+ -
Line 140: (mgt.PGMA-NH2)
Line 145: -NH2 groups
etc.
Pay attention to the location of the subscripts and superscripts all over the text.
Line 145: “were liked to alkyne groups”
Probably, “were linked to alkyne groups “
Line 218: pentaacetate acid (DTPA)
DTPA is Diethylene Triamine Pentaacetic Acid. Pentaacetate means the ionic form of the pentaacetic acid.
Line 405: αϒβ3 integrin,
αVβ3 integrin
--------------------------------
Main advantage to be noted: the presented review meets the recommendations of the Preferred Reporting Items for Systematic reviews and Meta-Analyses (PRISMA) guidelines and Quality Assessment of Diagnostic Accuracy Studies-2 (QUADAS 2) tools.
Author Response
I would thank you the revisor for the precious advice, in particular:
- The title has been modified to “Radiolabeled dendrimer coated nanoparticles for radionuclide imaging and therapy: a systematic review”;
- Line 21: the time interval has been corrected with January 1999 and reconciled with the sentence in line 72
- subscripts and superscripts have been corrected
- line 149: the sentence has been corrected with “were linked to alkyne groups “
- line 221: pentaacetate acid has been substituted with Diethylene Triamine Pentaacetic Acid
- line 409 and 412: the subscripts have been corrected (αϒβ3 integrin with αVβ3integrin)
- line 135: a description of the general structure and synthesis of dendrimers has been added.
Reviewer 2 Report
Authors did literture research and discussed appplications of dendrimers in 33 articles. In the discussion section, the manuscript was just piled up these articles, and did not organize nor discuss well. I suggest authors completely rewrite the discussion section clearly with a good structure and a logical flow. It would be more readable, if authors could provide some scheme or figures about the discussed papers.
Author Response
I would thank you the revisor for the precious advice. The discussion has been modified and better structured. Figures 4, 5, and 6 have been added.
Reviewer 3 Report
Thank you for the opportunity to review this article.
The article reviewed the application of dendrimers in the delivery of radionuclide for imaging of cancer. It is a good review that could give useful information on the area described.
However, the manuscript need a thorough English editing to improve understanding by the readers.
Table 1 and Table 2 may also be unnecessary as the information can be summarised in sentences and the table shows no comparative information between the items described.
Author Response
I would thank you the revisor for the precious advice. Table 1 and table 2 have been removed and English editing has been done.
Round 2
Reviewer 1 Report
I have no significant comments.
Reviewer 2 Report
none
Reviewer 3 Report
In my opinion, the manuscript is good to go.